# Numerical Simulation Study on the Relationships between Mineralized Structures and Induced Polarization Properties of Seafloor Polymetallic Sulfide Rocks

Caowei Wu [1,2], Changchun Zou [1,2,*], Cheng Peng [1,2,*], Yang Liu [1,2], Tao Wu [3], Jianping Zhou [3] and Chunhui Tao [3]

1   School of Geophysics and Information Technology, China University of Geosciences, Beijing 100083, China
2   National Engineering Research Center of Offshore Oil and Gas Exploration, Beijing 100028, China
3   Second Institute of Oceanography, Ministry of Natural Resources, Hangzhou 310012, China
*   Correspondence: zoucc@cugb.edu.cn (C.Z.); pengc@cugb.edu.cn (C.P.)

**Abstract:** The induced polarization (IP) method plays an important role in the detection of seafloor polymetallic sulfide deposits. Numerical simulations based on the Poisson–Nernst–Planck equation and the Maxwell equation were performed. The effects of mineralized structures on the IP and electrical conductivity properties of seafloor sulfide-bearing rocks were investigated. The results show that total chargeability increases linearly as the volume content of disseminated metal sulfides increases when the volume content is below 20%. However, total chargeability increases nonlinearly with increasing volume content in vein and massive metal sulfides when the volume content is below 30%. The electrical resistivity of disseminated metal sulfides mainly depends on the conductivity of pore water. The electrical resistivity of vein and massive sulfides mainly depends on the volume content and the length of sulfides. Increase in the aspect ratio (0.36 to 0.93) of seafloor massive sulfides causes relaxation time constants and total chargeability to decrease. Relaxation time constants and total chargeability also decrease with increase in the tortuosity of seafloor vein sulfides from 1.0 to 1.38. This study is of great value for the electrical survey of seafloor polymetallic sulfide deposits.

**Keywords:** seafloor polymetallic sulfide; mineralized structures; numerical simulation; induced polarization; Poisson–Nernst–Planck equation

## 1. Introduction

The exploitation of some deep, previously unprofitable seafloor ore resources has become economically viable with the advancement of marine-engineering technologies. The products of seafloor hydrothermal activities, seafloor polymetallic sulfides, widely occur in submarine structural regions, such as mid-ocean ridges and island arcs. Seafloor polymetallic sulfides are often associated with enriched metal resources with very high mining values [1,2]. The major metal sulfide minerals in seafloor sulfide deposits include pyrite (Fe-sulfide), chalcopyrite (Cu-sulfide), and sphalerite (Zn-sulfide). As of 2011, at least 165 potential seafloor polymetallic sulfide deposits have been discovered in the international seabed area, attracting the attention of countries and major economies across the world [3,4]. The Solwara project was conducted to investigate a seafloor polymetallic sulfide deposit located in the Bismarck Sea by Nautilus Minerals, Inc., which first publicly reported a seafloor mineral resource. In this project, multiple techniques have been adapted for the exploration, including remotely operated-vehicle video-dives, bathymetric surveys, and geophysical techniques. Geophysical techniques, especially electrical methods, provide non-intrusive and effective ways of detecting seafloor sulfide deposits [5]. For instance, self-potential and electromagnetic surveys were successfully carried out as part of the Solwara project. A detailed report was published in 2012 [6]. Self-potential and electromagnetic surveys refer to the redox potential and conductivity of metal sulfides, respectively [7].

Induced polarization (IP) is also an important electrical method. This method corresponds to the reversible charge storage of metal sulfide minerals. In general, there is an order-of-magnitude difference between the IP magnitudes of metal sulfides and their unmineralized host rocks [8]. Induced polarization measurement can be performed in both the time domain and the frequency domain. In general, frequency-domain induced polarization (FDIP) is often carried out in laboratory environments. Time-domain induced polarization (TDIP) is more cost-efficient and convenient for field investigation. The results of TDIP and FDIP measurements are almost equivalent [9]. Induced polarization was not used in the Solwara project, though there have been successful applications in other investigations of seafloor polymetallic sulfide deposits. A high IP effect has been shown in seafloor sulfide deposits and surrounding areas [10].

Compared with the self-potential and electromagnetic methods, the IP method provides more parameters for the detection of seafloor sulfide bodies. One of the key parameters in IP measurement is called chargeability, which describes the IP magnitude of rocks. Recently, laboratory IP measurements of seafloor drill cores have shown that there is a linear positive relationship between the chargeabilities and volume contents of disseminated metal sulfide minerals [11,12]. The above observations are similar to the experimental study results for terrestrial sulfide-bearing rocks. It is telling that the IP properties of seafloor polymetallic sulfides are similar to those of terrestrial metal sulfides [13,14]. Relaxation time distributions (RTDs) are also useful for interpreting TDIP data. A previous study on synthetic core samples proved that the size of metal sulfides can be roughly estimated from RTDs [15]. That said, the IP method contributes not only to delineating seafloor sulfide bodies but also to estimating the contents of metal sulfides.

Seafloor polymetallic sulfides, such as disseminated, massive, and vein sulfides, have developed in bedrock in various mineralized structures [16]. Massive and semi-massive sulfides with various aspect ratios mainly occur in the upper parts of deposits. Vein sulfides with different tortuosities can be observed in the lower parts of deposits and are often associated with disseminated sulfides. Previous research has shown that the total chargeability of disseminated sulfide rocks does not depend on the shape of sulfide particles [17,18]. The total chargeability of rocks containing sheet-like metal sulfides is less affected by their size and orientation relative to the direction of the polarizing field. The above dependence in rocks containing rod-like sulfides is much stronger [19–22]. However, studies on the effects of mineralized structures on the IP properties of seafloor polymetallic sulfides are currently lacking.

Natural seafloor metal sulfide rock samples are strongly anisotropic and very complicated. Collecting terrestrial metal sulfide ores with various aspect ratios and tortuosities to prepare synthetic rock samples is also difficult. Therefore, it is hard to investigate the effect of mineralized structures on the IP properties of seafloor polymetallic sulfides through laboratory experiments. Numerical simulations carried out according to the finite-element method are now regarded as useful tools with which to obtain good physical understanding of the IP mechanisms of conducting media [23]. Recently, a 2D numerical simulation based on the Poisson–Nernst–Planck equation has been widely used to reproduce the dependence of the IP properties of sand containing electrically conductive inclusions, such as the dependence of the relaxation time constant on the gain radius of conductive particles, background conductivity, and temperature [24,25], and the dependence of total chargeability on the volume contents of metallic particles [26]. The results obtained from numerical simulations are consistent with both the theory and the experimental data. Due to the cylindrical symmetry in the IP response of metal sulfides, numerical simulation only needs to be carried out based on 2D models [27].

This study aimed to investigate the effects of the mineralized structures of metal sulfides on the IP properties of seafloor sulfide-bearing rocks. TDIP parameters and resistivities of a series of numerical models representing seafloor sulfide-bearing rocks were obtained through finite-element numerical simulation. The differences in IP and conductivity properties between metal sulfides with different distributions—disseminated, massive,

and vein sulfides—were investigated. The shapes of massive and vein sulfides were described in terms of aspect ratios and tortuosities, respectively. After these subjects, the effects of these two parameters on the IP properties of metal sulfide rocks will be discussed.

## 2. Methods

### 2.1. Mineralized Structures of Seafloor Metal Sulfide Rocks

IP measurement can be carried out in seafloor boreholes through geophysical logging. As shown in Figure 1a, an IP meter can be placed at the bottom of a borehole after a seabed drilling operation. In situ and continuous IP information about the layer can then be recorded by pulling up the meter to the top of the borehole. As mentioned previously, the mineralized structures of sulfides developed in layers at different depths might also be different. This study concentrates on the effects of mineralized structures on the IP properties of seafloor sulfide-bearing rocks from three aspects. The first is the effect of the distribution of metal sulfides. Three sulfide distributions have been considered for this purpose, namely, massive metal sulfides, disseminated metal sulfides, and vein metal sulfides (Figure 1b). The second aspect is that of the aspect ratio of massive metal sulfides, which refers to the ratio between the long axis and the short axis of massive sulfides. The third aspect is that of the tortuosity of vein metal sulfides, which refers to the ratio of the length of vein sulfides to the distance between their ends.

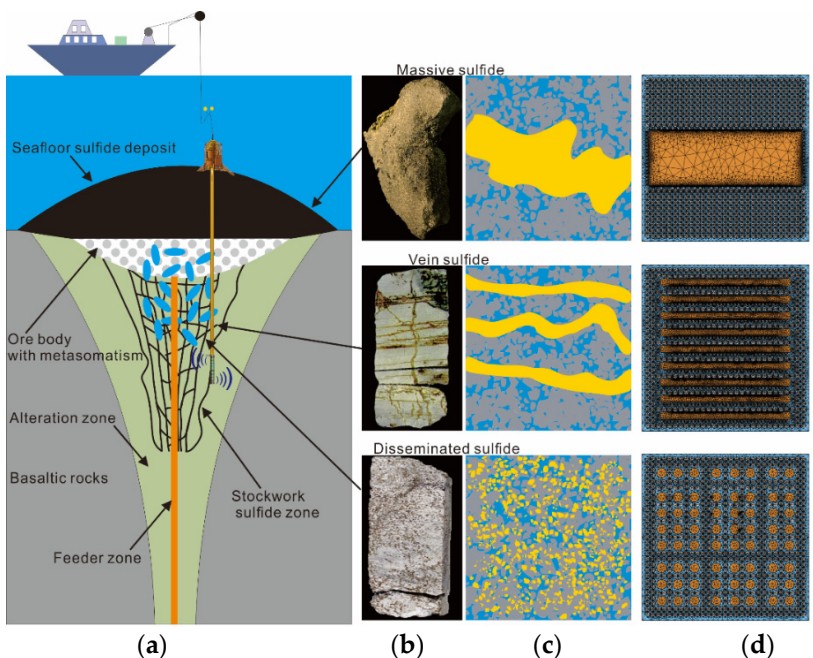

**Figure 1.** Representative mineralization structures in a seafloor polymetallic sulfide deposit. (**a**) Cross-sectional view of the distribution of the mineralized zones of a seafloor polymetallic sulfide deposit, modified from Fouquet et al. (2013) [28]. (**b**) Photographs of drill cores from seafloor sulfide deposits corresponding to the three kinds of mineralization. The photographs of the drill cores are from Marques et al. (2007) [29], Zierenberg et al. (1998) [30], and Anderson et al. (2019) [31]. (**c**) Schematic of three representative mineralized sulfide-bearing rocks. (**d**) Numerical models of seafloor sulfide-bearing rocks with various mineralized structures together with the finite-element mesh for the 2D numerical simulations. The grey circles denote basalt, the yellow zones denote metal sulfides, and the blue zones denote NaCl solution with a salinity close to that of seawater.

### 2.2. Numerical Simulation for the TDIP Response of Seafloor Metal Sulfide Rocks

The IP response of a seafloor polymetallic sulfide rock is the superposition of the IP response of the metal sulfides and the background porous material. The background porous material consists mainly of oceanic basalt and hydrothermal clay minerals. The IP of the background porous material is dominated by the electrical double layer formed

completely by ions in the mineral–fluid interface. The IP effect of a basaltic grain or clay is shown in Figure 2b; the interface of basalt or clay grain is coated with an electrical double layer (EDL). The inner part of the EDL is called the stern layer, while the outer part is called the diffuse layer [23]. Under the influence of an external electrical field, a dipole moment is created and associated with the transfer of the cations in the stern and diffuse layers. The direction of the dipole moment is opposite to that of the applied current.

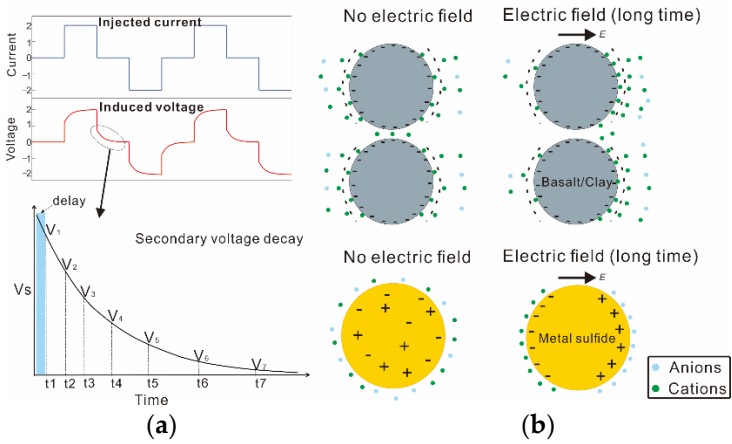

**Figure 2.** Sketch of TDIP measurement and the IP phenomenon. (**a**) Time-domain induced polarization measurement. (**b**) Induced polarization effect of a metal sulfide grain and basaltic/clay grain in the electrolyte.

Metal sulfide particles immersed in seawater are polarized under the influence of an applied electrical field. This phenomenon is caused by the migration and accumulation of charge carriers (electrons and electron holes). For nano- and microscale metal sulfide particles (<100 μm), polarization is more associated with the migration and accumulation of charge carriers along the surfaces of the metallic grains [27]. For larger-sized particles, the polarization is more associated with the migration and accumulation of charge carriers inside the metallic grains [18]. The IP effect of sulfide particles is shown in Figure 2b. The inside of a sulfide particle is electrically neutral, and the concentration of electrons and electron holes are the same everywhere when there is no electric field. The electrons move to one side of the particle and the electron holes move to the other side under an imposed electrical field. After the shutdown of the injection current, the electrons and electron holes diffuse/drift in the sulfide particle over time. A diffusion current and a corresponding secondary voltage that gradually decays to zero also appear. A metal sulfide is a good conductor in a high-frequency electric field, with perfect insulation in a low-frequency electric field due to the IP phenomenon. The critical frequency for the IP effect of disseminated metal sulfide particles is higher than that for massive and vein metal sulfides. High-salinity environments, such as the seafloor, also increase the critical frequencies of metal sulfides. According to a previous experimental study, the IP magnitudes of metal sulfides are an order of magnitude higher than those of the background [15]. In addition, the IP magnitudes of the background materials of seafloor sulfide-bearing rocks in high-salinity environments are lower than those in low-salinity environments [12]. Therefore, the IP responses of seafloor rocks containing metallic minerals are mainly controlled by the metallic minerals. The IPs of background materials can be neglected to simplify the numerical simulations in this study.

The signals measured in TDIP measurement using a four-electrode array are shown in Figure 2a. A bipolar periodic square current is applied to the material for some time and then shut down. The voltage at the two voltage electrodes, M and N, first decreases instantaneously and decays slowly. After an initial delay period, T', the secondary voltage decay is monitored, with distinct, different time intervals separated by characteristic times ($t_1$, $t_2$, $t_3$, etc.). The secondary voltage decay curves are recorded after the current is turned

off and converted to time-dependent chargeability (normalized secondary voltage decay curves) using the following equation:

$$m_i = \frac{1}{V_0(t_{i+1} - t_i)} \int_{t_i}^{t_{i+1}} V_s dt \tag{1}$$

where $V_0$ denotes the voltage just before the shutoff of the injection current, $V_s$ is the secondary voltage measured just after the shutoff of the injection current, and $m_i$ is the partial chargeability calculated from the time window $i$ between $t_i$ and $t_{i+1}$.

The dynamic change in the concentration of electrons/electron holes and the effect of the electric field on the flux of charge carriers during the induced polarization of the metal sulfides can be numerically simulated by the solution of Poisson–Nernst–Planck (PNP) equations. Poisson's equation describes the electric field inside and outside the metal sulfides. The Nernst–Planck equations describe the dynamic distribution of charge carriers in the metal sulfides with an external electric field. The Poisson–Nernst–Planck equation is written as follows:

$$\begin{cases} \frac{\partial c_i}{\partial t} = \nabla\left(D_i \nabla_{C_i} + \frac{D_i Z_i e}{k_B T} c_i \nabla_\psi\right); & i = 2 \\ \nabla(\varepsilon \nabla \psi) + \sum_i z_i e c_i = 0 \end{cases} \tag{2}$$

where $c_i$ is the concentration of each charge carrier (electron, electron hole, and ion), $z_i$ is the valence of each charge carrier, $\varepsilon$ is the dielectric permittivity, $\psi$ is electric potential, and $N$ is the number of involved charge carriers.

The numerical solution of the above-mentioned systems in the time domain (direct voltage during the current injection $T_{on}$ and no voltage during $T_{off}$) can be calculated using the finite-element method. A constant potential was imposed on the right and left sides of the models by means of a Dirichlet condition, which is written as follows:

$$\begin{cases} \psi_{left} = \frac{-\psi_0}{2} \\ \psi_{right} = \frac{\psi_0}{2} \end{cases} \tag{3}$$

It was also assumed that no flux of charge was allowed out of the top and bottom of the models where a Neumann boundary was applied. The Neumann boundary is written as follows:

$$\frac{\partial \psi}{\partial n} = 0 \tag{4}$$

The flow chart of the numerical simulation process based on the Poisson–Nernst–Planck equation is shown in Figure 3. Simplified 2D geometric models (Figure 1d) were built based on the three types of mineralization structures of seafloor sulfides mentioned above. We also changed the volume contents, aspect ratios, and tortuosities of the sulfides to make more geometric models. Finally, 25 simplified models with various distributions, aspect ratios, and tortuosities of metal sulfides were built. The porosity value for all of the models was 30%. Summaries of the characteristics and properties of these models are presented in Tables 1–3. Point electrodes for the voltage measurement were located close to the metal sulfides. For the sake of simplification, there were only three media used in the numerical simulation: seawater, metal sulfides, and basalt. The basalt was formed in a circle with a diameter of 1 mm. There were only four kinds of charge carriers in these models. Seawater only includes two kinds of ions, $Na^+$ and $Cl^-$, with a concentration of 500 mol/L. Metal sulfides only include electrons and electron holes, with a concentration of $6.02 \times 10^{23}/m^3$. The diffusion coefficient of the $Na^+$ and $Cl^-$ was $1.7 \times 10^{-9}$ m$^2$/s, while that of the electrons and electron holes was $2.7 \times 10^{-5}$ m$^2$/s. No oxidation–reduction reactions at the interface between the metallic particles and the background materials were considered. Hence, there was no exchange of electrons between sulfides and seawater. A blocking boundary condition was imposed at the interface of the metal sulfides and the NaCl solution. Basalt was regarded as a non-polarized material and included no-charge

carriers. Therefore, the solution of the PNP equations only reproduced the IP effects of metal sulfides.

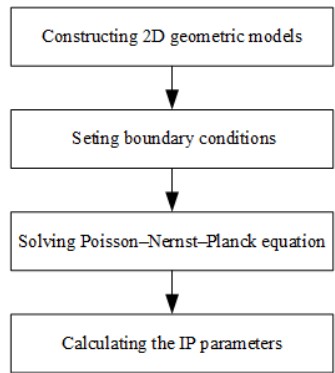

**Figure 3.** Flow chart of the numerical simulation process based on the Poisson–Nernst–Planck equation.

**Table 1.** Summary of the models corresponding to seafloor disseminated sulfide rocks. φ denotes the volume contents of metal sulfides; r denotes the grain radii of sulfide particles.

| No. | Φ (a.u.) | r (mm) |
|---|---|---|
| 1 | 0.04 | 1 |
| 2 | 0.08 | 1 |
| 3 | 0.15 | 1 |
| 4 | 0.12 | 1 |
| 5 | 0.20 | 1 |
| 6 | 0.20 | 1 and 4 |

**Table 2.** Summary of the models corresponding to seafloor vein sulfide rocks. L1 denotes the distances between the ends of vein sulfides; L2 denotes the lengths of the long axes of vein sulfides.

| No | φ | L1 (mm) | L2 (mm) | Tortuosity |
|---|---|---|---|---|
| 1 | 0.20 | 25 | 25.00 | 1.00 |
| 2 | 0.20 | 25 | 25.50 | 1.02 |
| 3 | 0.20 | 25 | 26.25 | 1.05 |
| 4 | 0.20 | 25 | 27.25 | 1.09 |
| 5 | 0.20 | 25 | 28.75 | 1.15 |
| 6 | 0.20 | 25 | 31.00 | 1.24 |
| 7 | 0.20 | 25 | 34.50 | 1.38 |
| 8 | 0.10 | 25 | 25.00 | 1.00 |
| 9 | 0.30 | 25 | 25.00 | 1.00 |

**Table 3.** Summary of the models corresponding to seafloor massive sulfide rocks. a denotes the length of the minor axes of massive sulfides; b denotes the lengths of the long axes of massive sulfides; α denotes the aspect ratios of massive sulfides.

| No. | φ | a (mm) | b (mm) | α |
|---|---|---|---|---|
| 1 | 0.20 | 8.75 | 24.00 | 0.36 |
| 2 | 0.20 | 10.00 | 21.00 | 0.48 |
| 3 | 0.20 | 11.05 | 19.00 | 0.58 |
| 4 | 0.20 | 11.67 | 18.00 | 0.65 |
| 5 | 0.20 | 12.35 | 17.00 | 0.73 |
| 6 | 0.20 | 13.13 | 16.00 | 0.82 |
| 7 | 0.20 | 14.00 | 15.00 | 0.93 |
| 8 | 0.10 | 6.19 | 16.97 | 0.36 |
| 9 | 0.30 | 10.72 | 29.39 | 0.36 |
| 10 | - | | | - |

To increase the accuracy of the simulation, a finer mesh was used inside the sulfides, where the concentration of charge carriers changes quickly and the polarization phenomenon takes place, while a coarser mesh was used for the basalt and pore water to accelerate computation speed.

The initial conditions of the metal sulfide models before the numerical simulation (t = 0) were set as follows: (1) the electric potential was 0 everywhere in the metal sulfide models; (2) the concentrations of electrons and electron holes within the metal sulfides were equal and had uniform distributions; (3) the concentrations of $Na^+$ and $Cl^-$ were the same everywhere in the electrolyte. The Dirichlet boundary condition was set in the right and left boundaries. Therefore, the electric field was always along the horizontal direction and parallel to the long axis of the metal sulfides.

Electrical resistivity, also, is regarded as an important indicator in the detection of seafloor polymetallic sulfide deposits. Low-frequency resistivity is normally measured in situ through TDIP or another direct-current (DC) electric method. High-frequency resistivity can be measured by FDIP and the inductive electromagnetic method. Metal sulfides behave as insulators under low-frequency currents and as good conductors under high-frequency currents due to the IP effect. A previous study showed that low-frequency resistivity is more sensitive to ionic conduction through seawater in the pores and throats of rocks rather than the electron conduction of disseminated metal sulfide particles. Therefore, this study focused on the influence of sulfide distribution on high-frequency resistivity rather than low-frequency resistivity. Numerical simulations of the electrical conductive properties of seafloor sulfide-bearing rocks were also performed based on the Maxwell equation. The electrical conductivity of the metal sulfides was 1000 S/m and that of the basalt was 0 in the numerical simulation and that of the seawater was 4 S/m.

The Maxwell equation for a constant electric field is written as follows:

$$\begin{cases} \nabla \cdot E = \frac{\rho}{\varepsilon_0} \\ \nabla \times E = 0 \end{cases} \tag{5}$$

where $E$ is the electric field intensity in $V \cdot m^{-1}$, $\rho$ is the electric density in $C \cdot m^{-1}$, and $\varepsilon_0$ is the dielectric constant in $F \cdot m^{-1}$.

Based on the law of electric charge conservation, the electric-current density of a constant electric field satisfies the equation written as follows:

$$\nabla \cdot J = 0 \tag{6}$$

### 2.3. Calculation of the Induced Polarization Parameters

Based on Ohm's law, the resistivity, $R$, of the metal sulfide models is written as follows:

$$R = rK = K \frac{U_1 - U_0}{I} \tag{7}$$

where $U_0$ and $U_1$ are the electric potentials at both ends of the metal sulfide models and $I$ is the electric current in $A$. The $K$ factor is written as follows:

$$K = \frac{S}{L} \tag{8}$$

where $S$ is the cross-sectional area of the models (perpendicular to the electric field) and $L$ is the length of the models (parallel to the direction of voltage application). In this numerical simulation, the direction of the electric field is parallel to the $x$-axis, so the $K$ factor is written as follows:

$$K = \frac{L_y L_z}{L_x} \tag{9}$$

The electric current, *I*, can be calculated by obtaining the integral of the normal current density of the input or output voltage.

$$I = \iint_S J_n \cdot dS \tag{10}$$

where *n* is the direction of the injection current, which is in the *x*-direction in this numerical simulation. The resistivity, *R*, can be calculated by substituting Equations (9) and (10) into Equation (7).

The secondary voltage decay curve obtained from the numerical simulation can be inverted to determine the Cole–Cole parameters and the relaxation distribution, which are known to be able to adequately describe the IP properties of rocks.

The most extensively used model in the interpretation of TDIP data is the Cole–Cole model. The Cole–Cole model can be written in the time domain to find a suitable expression for the secondary voltage decay curve, which is given by Guptasarma (1982) [32].

$$\begin{cases} m(t) = M \left[ \sum_{n=0}^{\infty} \frac{(-1)^n \left(\frac{t}{\tau_0}\right)^{nc}}{\Gamma(1+nc)} \right] \left(0 \le \frac{t}{\tau_0} \le 2\pi\right) \\ m(t) = M \left[ \sum_{n=1}^{\infty} \frac{(-1)^{n+1} \left(\frac{t}{\tau_0}\right)^{-nc}}{\Gamma(1-nc)} \right] \left(\frac{t}{\tau_0} > 2\pi\right) \end{cases} \tag{11}$$

where *c* is the Cole–Cole exponent (dimensionless), $\tau_0$ denotes the Cole–Cole time constant (in *s*), and *M* denotes the total chargeability (dimensionless). Γ denotes the Euler gamma function, defined as:

$$\Gamma(x) = \int_0^{\infty} u^{x-1} e^{-u} du \tag{12}$$

The time-domain Cole–Cole model is used to fit the measured secondary voltage decay curves based on the simulated annealing method. The minimization of the objective function is written as:

$$\psi(M, c, \tau) = \sqrt{\frac{1}{N} \sum_{i=1}^{N} \left( m_{obs}(t_i) - m_{fit}(t_i) \right)^2} \tag{13}$$

where $m_{obs}(t_i)$ is the apparent chargeability at a given time $t_i$, $m_{fit}$ is the fitted apparent chargeability at time $t_i$, and *N* is the number of sampling points.

A relaxation time distribution (RTD) can characterize polarization magnitude as a function of its characteristic relaxation time. These characteristic relaxation times relate to the physical properties of rocks, such as pore-size distribution in sandstone, and the polarizable sources [33–35]. In this study, based on the time-dependent chargeability decays obtained from the numerical simulation, the RTD was inverted using a regularization method. This method combines the Cole–Cole model, the conventional singular-value decomposition inversion method, and regularization constraints to derive smooth and continuous RTDs [36]. Moreover, the derived RTDs are little affected by the signal-to-noise ratio of IP decay. The process of RTD inversion is detailed in Appendix A.

## 3. Results

### 3.1. TDIP Parameters of the Models with Different Mineralized Structures

The dependence of total chargeability on the volume content of metal sulfides is shown in Figure 4. The numerical simulation results were also compared with the previous experimental data for natural or synthetic seafloor sulfide rocks. The total chargeability of disseminated sulfides increased linearly from 0.19 to 0.89 by increasing the volume content of metal sulfides from 4% to 20%. It should be mentioned that the total chargeability was 0 when the volume content of sulfides was 0 because the background material was non-polarized in our numerical simulation. The linear relationship between the chargeability

and the volume content of the disseminated sulfides was also consistent with both the experimental data and the prediction proposed by Revil et al. (2015) [18]. Therefore, the results obtained from the numerical simulation are valid. However, there was no linear correlation found between total chargeability and volume content in the vein or massive sulfides.

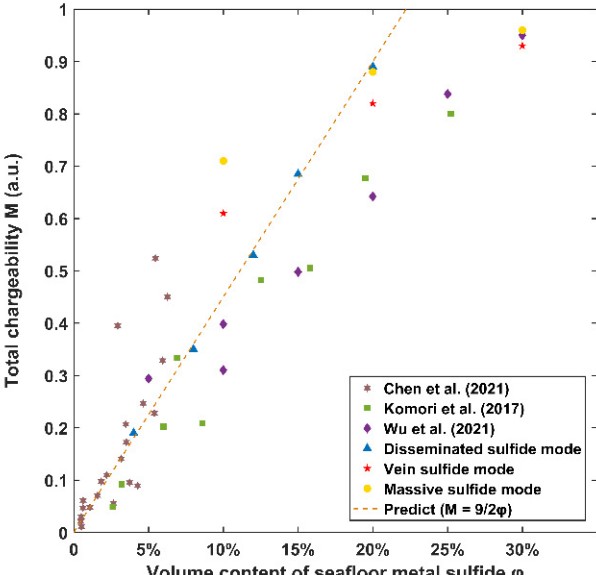

**Figure 4.** The relationship between chargeability and volume content in seafloor metal sulfides. The simulation data (vein sulfide models, massive sulfide models, and disseminated sulfide models) are from the present study. The experimental data for natural seafloor sulfide rocks are from Chen et al. (2021) [37] and Komori et al. (2017) [10]. The type of sulfides used in Chen et al. (2021) was disseminated and the Komori study included disseminated and massive sulfides. The experimental data for synthetic seafloor sulfide rocks are from Wu et al. (2021) [15]. The red line denotes the prediction of the models proposed by Revil et al. (2015) [18]. In this model, the value of total chargeability was equal to 4.5 times the value of the volume content of disseminated metal sulfides.

The total chargeabilities and relaxation time constants of seafloor massive sulfide rocks with various aspect ratios are shown in Figure 5a. The total chargeabilities decreased from 0.68 to 0.35 and the relaxation time constants decreased from 0.2 s to 0.018 s by increasing the aspect ratios of seafloor massive sulfides from 0.36 to 0.93. The total chargeabilities and relaxation time constants for the vein metal sulfide models with different tortuosities are shown in Figure 5b. The total chargeabilities decreased from 0.82 to 0.62 and the relaxation time constants decreased from 0.053 s to 0.030 s by increasing the tortuosities of seafloor vein sulfides from 1.00 to 1.38.

The relaxation time distributions for the disseminated metal sulfide models with only one sulfide grain radius obtained only one distinct peak each (Figure 6). The peak relaxation time for disseminated sulfides with a grain radius of 1 mm was 0.33 ms and the amplitude of the peak was 0.0058. The peak relaxation time for disseminated sulfides with a grain radius of 4 mm was 4.06 ms, and the amplitude of the peak was 0.0068. The larger the sulfide grain size, the higher the value of the peak relaxation time. The results are in agreement with those of previous experimental studies [13–15]. A bimodal relaxation time distribution was obtained from the disseminated sulfide models including sulfide particles with grain radii of 1 mm and 4 mm. The first peak was at 0.28 ms and the second peak was at 3.92 ms. The amplitude of the first peak was 0.0022 and that of the second was 0.0038.

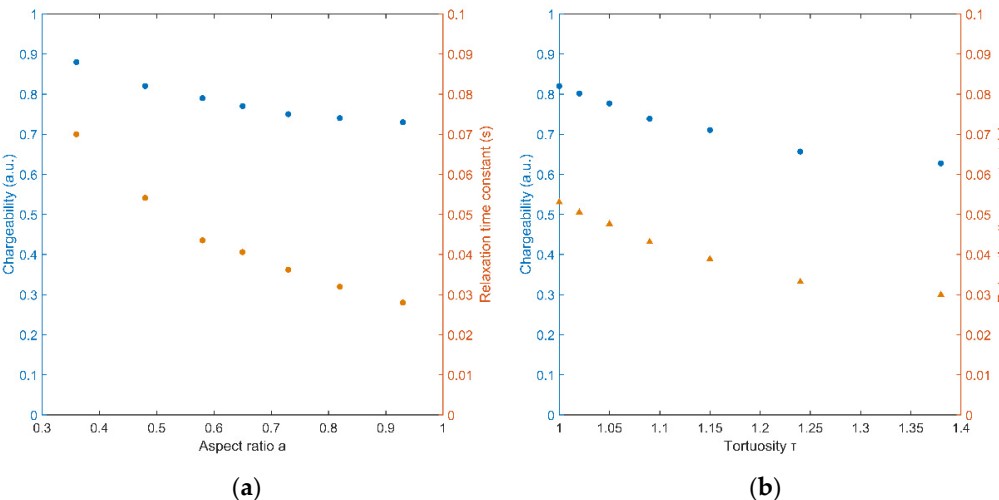

**(a)**                                                           **(b)**

**Figure 5.** Dependence of the total chargeabilities and relaxation time constants of seafloor sulfide-bearing rocks on the aspect ratios and tortuosities of metal sulfides. (**a**) The total chargeabilities, M, and the relaxation time constants, τ, for the seafloor massive metal sulfide models with various aspect ratios. (**b**) The total chargeabilities, M, and relaxation time constants, τ, for the seafloor vein metal sulfide models with various tortuosities.

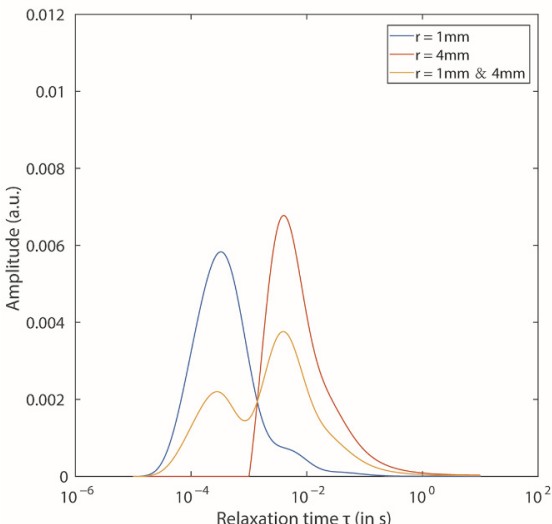

**Figure 6.** Relaxation time distributions for the seafloor disseminated metal sulfide models with sulfide particles with various grain radii. The blue line denotes models containing fine sulfide particles with a grain radius of 1 mm; the red line denotes models containing coarse sulfide particles with a grain radius of 4 mm; the yellow line denotes models containing both fine and coarse sulfide particles in equal proportions.

The RTDs for the seafloor massive sulfide models with different sulfide aspect ratios and the same sulfide volume content of 20% are shown in Figure 7. The amplitudes of the RTD peaks decrease from 0.0706 to 0.0329 and the relaxation times of the peaks decreased from 0.154 to 0.0121 s as the aspect ratios were increased from 0.36 to 0.93. The RTDs for the seafloor vein sulfide models with various sulfide tortuosities and the same sulfide volume content of 20% are shown in Figure 8b. The peaks of the RTDs change from 0.0142 to 0.0104 and the peak relaxation times change from 0.0554 to 0.0353 s with increasing tortuosities from 1 to 1.38. The relaxation time distributions for all the seafloor vein and massive sulfide models were unimodal, which means that the peak number was less affected by aspect ratio and tortuosity.

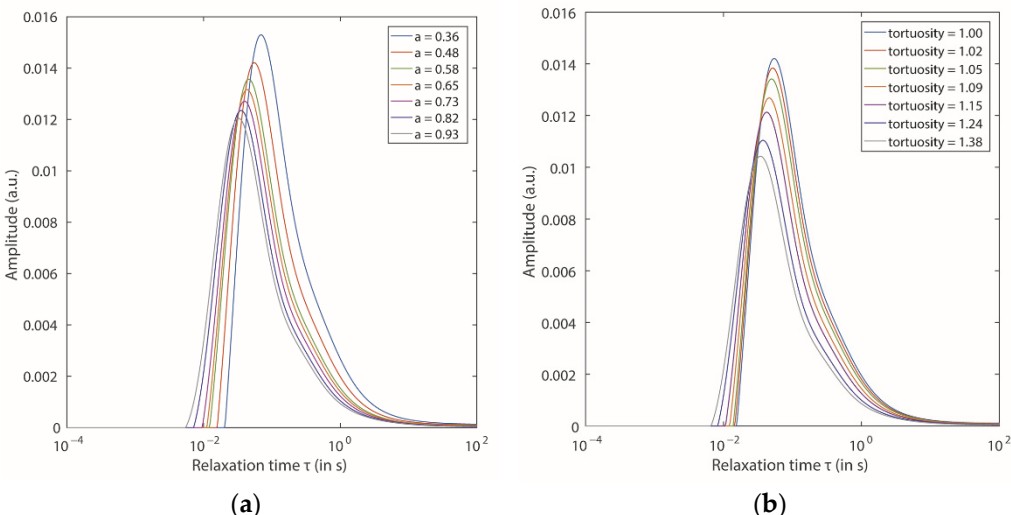

**Figure 7.** Dependence of the relaxation time distributions of seafloor sulfide-bearing rocks on the aspect ratios and tortuosities of metal sulfides. (**a**) The relaxation time distributions for seafloor massive metal sulfide models with various aspect ratios. (**b**) The relaxation time distributions for seafloor vein metal sulfide models with various tortuosities.

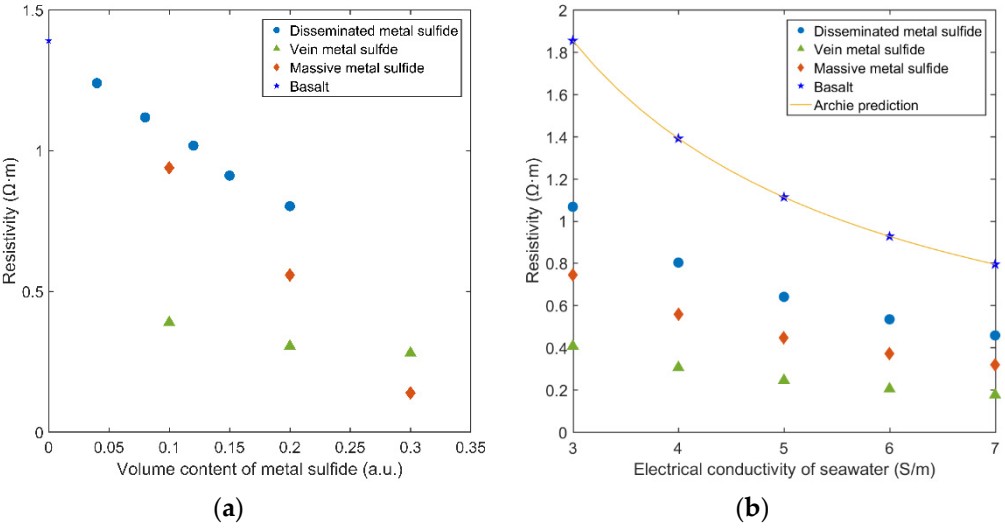

**Figure 8.** Dependence of the resistivities of seafloor sulfide-bearing rocks on the volume contents of metal sulfides and the electrical conductivity of seawater. (**a**) Resistivity versus volume content for the metal sulfides. (**b**) Resistivity versus electrical conductivity for seawater.

*3.2. Resistivities of the Models with Various Sulfide Contents and Electrical Conductivity of Seawater*

The resistivities of the seafloor polymetallic sulfide models with different mineralized structures and the same sulfide volume content of 0.2 were between 0.31 and 0.80 $\Omega \cdot$m. The resistivities of the massive and vein metal sulfide models were smaller than those of the disseminated metal sulfide models with the same volume content of metal sulfides. The resistivity of the disseminated sulfides decreased from 1.24 to 0.81 $\Omega \cdot$m as the volume content of metal sulfides was increased from 4% to 20%. The resistivity of the vein sulfide decreased from 0.39 to 0.29 $\Omega \cdot$m and that of the massive sulfide decreased from 0.94 to 0.14 $\Omega \cdot$m as the volume content of metal sulfide was increased from 10% to 30% (Figure 8a). The resistivity of the disseminated sulfide decreased from 1.07 to 0.46, the resistivity of the vein sulfide decreased from 0.41 to 0.18$\Omega \cdot$m, and that of the massive sulfide decreased from 0.74 to 0.32 $\Omega \cdot$m as the electrical conductivity of pore water was increased from 3 S/m to 7 S/m (Figure 8b).

## 4. Discussion

### 4.1. Effect of Sulfide Distribution on the TDIP Properties of Seafloor Sulfide-Bearing Rocks

This section started by discussing the differences in the M–φ relationship between seafloor disseminated, vein, and massive sulfides. The numerical simulation results showed that the dependence of total chargeability on the volume content of sulfides for the vein and massive sulfides was different from that for the disseminated metal sulfides. The total chargeabilities of vein and massive metal sulfides were higher than that of the disseminated metal sulfide with the same volume content of metal sulfides of 10%. However, the total chargeabilities of the vein and massive metal sulfides were lower than that of the disseminated metal sulfide when the volume content of metal sulfide was 20%. That said, chargeability was more sensitive to volume content in the vein and massive metal sulfides than in the disseminated metal sulfide when the volume content of metal sulfides was low. However, the dependence of total chargeability on volume content in the vein and massive metal sulfides decreased as the volume content of metal sulfides was increased. Total chargeability reflects the IP magnitude of sulfides, which is determined by the overvoltage at two sulfide particle poles [14]. With increase in grain radius, the magnitude of the overvoltage increases, which leads to a greater magnitude of polarization. Therefore, the IP magnitudes of the vein and massive metal sulfides were much larger than those of the disseminated sulfide particles when the volume content of sulfides was low. By increasing the volume content of metal sulfides, a complex conductive network is formed. More and more charge carriers inside the sulfides tend to electrical conduction rather than induced polarization. Therefore, a decrease in IP magnitude in the vein and massive sulfides resulted when the volume content of metal sulfides was high.

Both sets of RTDs for the vein and massive sulfide models presented only one distinct peak. The RTDs for the models with disseminated sulfides of the same grain radius also presented only one distinct peak. However, the peak relaxation times for the vein and massive sulfide models were much larger than those for the model containing disseminated sulfide. That said, the shapes of the RTDs for the seafloor metal sulfides with different mineralized structures were also different. The RTDs for the disseminated metal sulfide models with narrow size distributions of sulfide particles should have been unimodal with strong amplitudes and small peak relaxation times. The RTDs for disseminated metal sulfide models with broad grain size distributions could have been multimodal. The RTDs for the vein and massive metal sulfide models should have been unimodal with strong amplitudes and large peak relaxation times. The IP effects of these non-mineralized rocks, such as oceanic basalt (the bedrock of seafloor sulfide deposits), are weak and slow, as the IP effects of non-mineralized rocks are dominated by the electrical double layers formed completely by ions at the mineral–fluid interface. Hence, an RTD for bedrock may show a broad peak with a small amplitude and a large peak relaxation time. Bimodal RTDs were obtained for the models containing disseminated metal sulfides of two different grain sizes. The first peak with a smaller relaxation time was attributed to the sulfide particles with a grain radius of 1 mm. The second peak at 3.92 ms was interpreted as corresponding to the larger sulfide particles with a grain radius of 4 mm. Therefore, RTDs can show multiple peaks when there is a superposition of the polarization of sulfide minerals with different grain sizes or mineral structures causing the IP effect. It is indicated that RTDs can play an important role in the detection of the structures of seafloor sulfide deposits. The RTDs for the top of a seafloor sulfide deposit may be unimodal with strong amplitudes and large peak relaxation times due to the concentration of massive sulfides in domal bodies. The RTDs for deep seafloor sulfide deposits may present multiple peaks due to the development of sulfide stockwork composed of vein and disseminated metal sulfides.

### 4.2. Effect of Sulfide Distribution on the Electrical Conductive Properties of Seafloor Sulfide-Bearing Rocks

The numerical simulation results show that the resistivities of all the models were more than an order of magnitude below the prediction curves. The prediction curves represented

the relationships between the resistivities of rocks and pore water according to the Archie equation (Figure 8b). Resistivity increases as the volume content of metal sulfides increases. A slight increase in the electrical conductivity of pore water resulted in a large decrease in the resistivities of the disseminated metal sulfide models. In contrast, the resistivities of the disseminated sulfide models increased slightly as the volume content of sulfides increased. That said, the high-frequency resistivity of disseminated metal sulfides still seems to be more sensitive to changes in pore-fluid conductivity than the volume content of metal sulfides. The resistivities of the massive and vein sulfide models were much lower than in the disseminated sulfide models. This was because both massive and vein metal sulfides can form sufficiently long electronic conduction paths.

The current density distribution diagrams for each model are shown in Figure 9. Most of the current in the massive and vein metal sulfide models passed through the metal sulfides. Most of the current in the disseminated metal sulfides was still distributed in the pore water. That said, the main conductive path inside the seafloor rocks containing disseminated metal sulfides was the connected pore water. The electrical conductivity of connected pore water is usually determined by pore structure (pore size, pore-size distribution, and pore morphology). However, the effect of pore structure on the resistivity of seafloor polymetallic sulfide rocks was significantly minimized. This was due to the high porosity of sulfide-bearing rocks and pore water with very high conductivity.

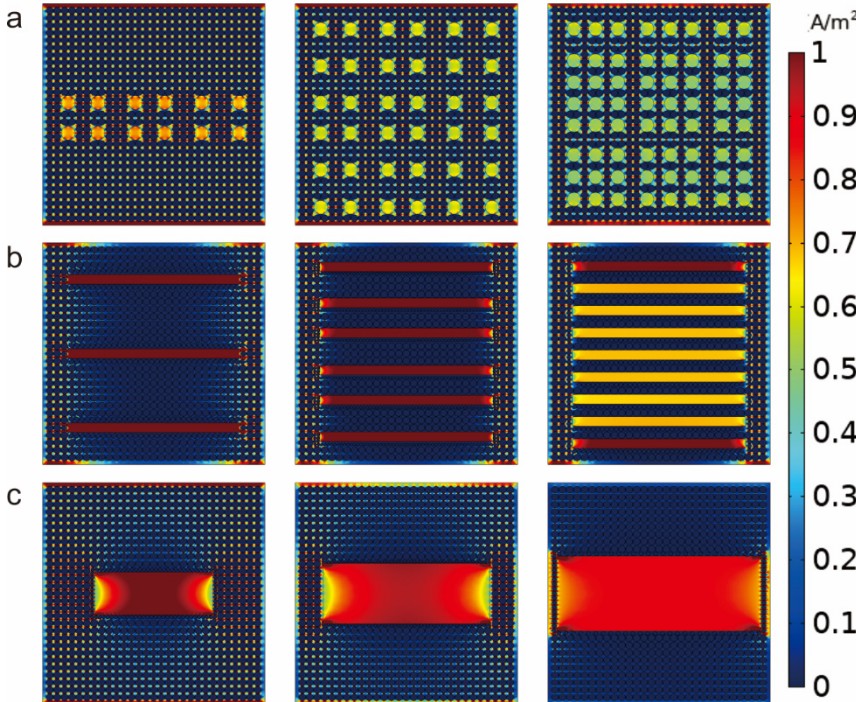

**Figure 9.** Electric-current density distributions of models of seafloor sulfide-bearing rocks under a horizontal electric field. (**a**) Disseminated sulfide models with various sulfide contents of 4%, 12%, and 20%. (**b**) Vein sulfide models with various sulfide contents of 10%, 20%, and 30%. (**c**) Massive sulfide models with various sulfide contents of 10%, 20%, and 30%. The red zones denote a high electric-current density, which means that more current flows through these areas.

The resistivity of massive metal sulfides decreased greatly by increasing the volume content of sulfides. The resistivity was more sensitive to volume content in the massive sulfides than in the disseminated sulfides. However, the resistivity of vein metal sulfides in this simulation seemed to be less affected by the volume content of metal sulfides. The length of massive sulfides parallel to the electric field increased with the increase in the volume content of metal sulfides in this numerical simulation. By contrast, the length of vein sulfides did not change with increasing volume content of metal sulfides in the

numerical simulation. It was indicated that the resistivities of the vein and massive sulfides were more sensitive to the length of sulfides parallel to the electric field than to the volume content of metal sulfides. In summary, the resistivities of massive and vein-like metal sulfides mainly depended on the length of the sulfides parallel to the electric field. The resistivity of the disseminated metal sulfide mainly depended on the electrical conductivity of the pore water.

### 4.3. Effects of Aspect Ratios and Tortuosities on the TDIP Properties of Seafloor Sulfide-Bearing Rocks

The numerical simulation results showed that the lower the aspect ratio parallel to the induced polarization field, the smaller the relaxation time constant and the total chargeability of massive sulfides. This was because both the overvoltage at two sulfide particle poles and the migration distance of the charge carriers inside the sulfides decreased with the decrease in the lengths of sulfides parallel to the induced polarization field. A previous study showed that the relaxation time constant was determined by the characteristic size of metal sulfides. For disseminated sulfides, the relaxation time constants were proportional to the squares of the grain radii of sulfide particles. For rod-like metal sulfides, the relaxation time constants were observed to be proportional to half of the major axes of the inclusion [18,19]. As shown in Figure 10, there was a clear linear positive correlation between the relaxation times and the length of massive metal sulfides parallel to the induced polarization field when the aspect ratio of the sulfides was small. When the aspect ratio was close to 1, the relaxation time constant seemed to be proportional to the square of the length of the massive sulfides. That said, the relationship between the characteristic length and relaxation time constants of massive sulfides changed with the aspect ratio.

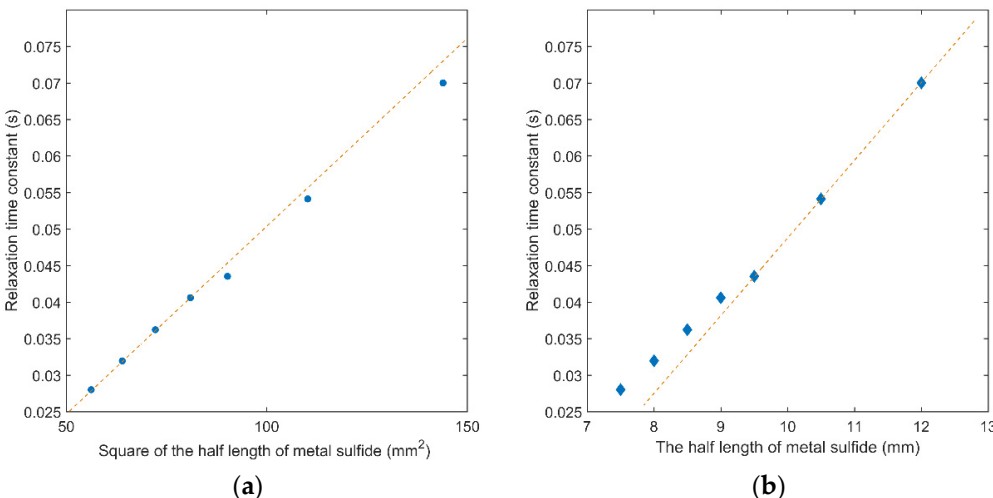

**Figure 10.** Dependence of relaxation time constants on the square of half of the length and half of the length of seafloor massive sulfides. (**a**) Relaxation time constants versus the squares of half of the lengths. (**b**) Relaxation time constants versus half of the lengths.

According to the numerical simulation results, increasing tortuosity decreased the total chargeability and relaxation time constants of vein metal sulfides. The decrease in the migration distance of the charge carriers inside the sulfides cannot be accounted for by the decrease in relaxation time constants. This is because the lengths of vein sulfides with various tortuosities in this study were the same. The concentration distributions of electrons inside the vein metal sulfide models with various tortuosities at the moment of cutoff are shown in Figure 11a. Most of the electrons accumulated at the sulfide–electrolyte interface when the tortuosities of the vein sulfides were close to 1. For the vein sulfides with high tortuosities, high electron concentration regions were hardly observed at the inner boundaries of the sulfides. It was indicated that a portion of the electrons within the vein metal sulfides with high tortuosities did not participate in the IP effects.

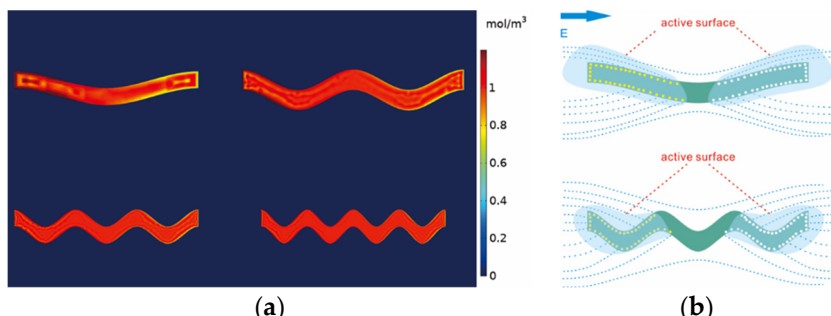

**Figure 11.** Influence of the tortuosities of vein metal sulfides on the migration of electrons during the induced polarization. (**a**) The numerical simulation results for the concentration distributions of electrons inside the vein metal sulfide models with various tortuosities just after the cutoff. (**b**) Schematic representation of the surface charge distributions around polarized vein metal sulfides with different tortuosities.

Gurin et al. (2021) [22] proposed an active-surface theory in the study of the IP properties of anisotropic rocks with metallic particles. According to this theory, the relaxation time constant is positively associated with the area of the active surface. When the induced polarization field is parallel to the metal mineral's long axis, the area of the active surface is larger than when the electric field is parallel to the metal mineral's minor direction. Therefore, the area of the active surface decreases as the lengths of vein sulfides parallel to the induced polarization field decrease, which causes a decrease in relaxation time. The effect of tortuosity on the induced polarization properties of sulfides can also be explained by active surface theory. In the case of seafloor metal sulfides with low tortuosities, most of the inner faces of sulfides can be regarded as active surfaces due to the distribution of electric potential. For vein sulfides with high tortuosities, only the surface close to the end of the sulfide is an active surface (Figure 11b). Therefore, increase in tortuosity leads to a decrease in the areas of the active surfaces of vein sulfides, which causes a decrease in relaxation time constants.

## 5. Conclusions

Finite-element numerical simulations were carried out to investigate the effects of mineralized structures on the IP and electrical conductive properties of seafloor sulfide-bearing rocks. The main observations and interpretations are as follows:

(1) Total chargeability increases linearly with volume content in disseminated sulfides when volume contents are below 20%. However, total chargeability depends nonlinearly on the volume contents of vein and massive metal sulfides. The distribution of sulfide minerals at different depths in a sulfide-rich body can be estimated from RTDs.

(2) The major conducting path for the electric current of seafloor disseminated metal sulfides is the connected pore, while that for vein and massive sulfides is via the metallic minerals with electrical conductivity in the electrical conductivity range of seawater, from 3.0 S/m to 7.0 S/m. Furthermore, electrical resistivity was found to be more sensitive to the lengths of vein and massive sulfides parallel to the electric field than to sulfide volume contents.

(3) The increase in aspect ratio leads to a decrease in the length of the dipole moment and the overvoltage on both sides of metal sulfides, which causes a decrease in the relaxation time constants and total chargeability. The areas of the active surfaces of sulfides decrease as the tortuosities of vein sulfides increase, and this leads to a decrease in relaxation time constants when the tortuosities are within the range of 1.0 to 1.38.

**Author Contributions:** Conceptualization, C.W.; methodology, C.W.; software, C.W. and Y.L.; validation, T.W.; formal analysis, C.W.; investigation, C.W.; resources, C.T. and J.Z.; data curation, C.W. and Y.L.; writing—original draft preparation, C.W.; writing—review and editing, C.Z. and C.P.; visualization, C.W.; supervision, C.Z. and C.P.; project administration, C.Z. and C.P.; funding acquisition, C.Z. and J.Z. All authors have read and agreed to the published version of the manuscript.

**Funding:** This research was funded by the National Natural Science Foundation of China (no. 41874215) and the COMRA Major Project under contract (no. DY135-S1-01-06).

**Data Availability Statement:** The data used to support the findings of the study are available from the first author upon request.

**Conflicts of Interest:** The authors declare no conflict of interest.

## Appendix A

The induced polarization decay of metallic particles can be expressed mathematically as a weighted superposition of a series of Cole–Cole exponential decays based on $\sum_{a=0}^{\infty} \frac{(-1)^a \left(\frac{t}{\tau_0}\right)^{ac}}{\Gamma(1+ac)}$ with different time constants, $\tau_i$, which relate to the individual polarization elements, and $c = 1$. Hence, the measured induced polarization decay curve can be described as follows:

$$
\begin{cases}
m(t) = \int_0^{\infty} \sum_{a=0}^{\infty} \frac{(-1)^a \left(\frac{t}{\tau_0}\right)^{a}}{\Gamma(1+a)} G(\tau) d\tau \ (0 \le \frac{t}{\tau_0} \le 2\pi) \\
m(t) = \int_0^{\infty} \sum_{a=1}^{\infty} \frac{(-1)^{a+1} \left(\frac{t}{\tau_0}\right)^{-a}}{\Gamma(1-a)} G(\tau) d\tau \ (\frac{t}{\tau_0} > 2\pi)
\end{cases}
\tag{A1}
$$

To calculate the relaxation time distribution, $G(\tau)$, Equation (A1) needs to be discretized as:

$$
Ax = b, \ A \in R^{n \times m}, x \in R^m, b \in R^n
\tag{A2}
$$

where:

$$
\begin{cases}
A_{n \times m} = \sum_{a=0}^{\infty} \frac{(-1)^a \left(\frac{t_n}{\tau_m}\right)^a}{\Gamma(1+a)} \ 0 \le \frac{t}{\tau_0} \le 2\pi \\
A_{n \times m} = \sum_{a=1}^{\infty} \frac{(-1)^{a+1} \left(\frac{t_n}{\tau_m}\right)^{-a}}{\Gamma(1-a)} \ \frac{t}{\tau_0} > 2\pi
\end{cases}
\tag{A3}
$$

$$
x_{m \times 1} = (G_1, G_2, \ldots\ldots G_m)^T
\tag{A4}
$$

where $A_{n \times m}$ denotes the matrix of the dimensional coefficient of $\sum_{a=0}^{\infty} \frac{(-1)^a \left(\frac{t}{\tau_0}\right)^{ac}}{\Gamma(1+ac)}$, $t_n$ is the moment of measuring the secondary voltage, $b_{n \times 1}$ denotes the column vector of the discrete secondary voltage data, $x_{m \times 1}$ denotes the column vector of the solution representing the $G(\tau)$ of the relaxation time constant $\tau_m$, and $\tau_m$ (m = 1, 2, 3 . . . , $i$) denote the series of relaxation time constants selected from $\tau_{min}$ to $\tau_{max}$ in advance.

Moreover, the solution vector, $x$, is sensitive to the noise of TDIP data; slight noise signals in the secondary voltage decay curves could cause large deviations in the inversion results. Therefore, Equation (A2) needs a special treatment to solve $x$. The new equation can be expressed as:

$$
\min : ||b - Ax||^2 + \lambda^2 ||Wx||^2
\tag{A5}
$$

where $W$ denotes the regularization operator and $\lambda$ denotes the regularization parameter. In this inversion algorithm for RTD, Tikhonov regularization was chosen as the regularization method, while the unit matrix was chosen as the regularization matrix and the $L$ curve method was chosen as the regularization-parameter determination method.

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
