# Peer review of "Numerical Simulation Study on the Relationships between Mineralized Structures and Induced Polarization Properties of Seafloor Polymetallic Sulfide Rocks"

_minerals, doi:10.3390/min12091172_

Round 1
Reviewer 1 Report
Dear editor,
I have reviewed the manuscript entitled “Relationships between mineralized structures and induced polarization properties of seafloor polymetallic sulfide rocks: A numerical simulation study”, in which the numerical simulations based on the Poisson–Nernst–Planck equation and Maxwell equation were performed to investigate the effect of mineralized structures on the induced polarization and electrical conductivity properties of seafloor sulfide-bearing rocks. Although the idea is of interesting, the structure and scientific discussions of the current version of the manuscript are to be improved for publication. Anyway, there are some comments which may be helpful for authors:
1) It is better to change the manuscript’s title by a clearer subject.
2) There are some spelling and grammatical errors, which dictate an in-depth review.
3) It is better to present some engineering findings and quantified data in Abstract.
4) The literature survey needs to be improved by other up-to-date researches.
5) The theory and methodology sections are not comprehensive, and need more discussions based on the reliable references.
6) It is suggested to present a flow chart or a pseudo code for better understanding the modeling process.
7) Fig. 5 needs more clarifications.
8) How could you optimize the numerical simulations?
9) Did the authors evaluate the validity of their obtained results?
Kind regards,
Author Response
I am very grateful to your comments for the manuscript. Those comments are all valuable and very helpful for revising and improving our paper, as well as the important guiding significance to our researches. According to your comments above, we have made extensive modification on the original manuscript. The point-by-point response to the the reviewer’s comments is in the attachment.

Reviewer 2 Report
The manuscript entitled “Relationships between mineralized structures and induced polarization properties of seafloor polymetallic sulfide rocks: A numerical simulation study” by Wu et al. performed numerical simulations based on Poisson-Nernst-Planck equation and Maxwell equation to study the effect of mineralized structures on the induced polarization and electrical conductivity properties of seafloor sulfide-bearing rocks. The results are clear and creative. However, there are many gramma mistakes, and language editing by a senior professor is needed.
Author Response
I am very grateful to your comments for the manuscript. We carefully revised the paper to make the English more fluent and correct. We also revised these places where the sentences long and complex. These changes will not influence the content and framework of the paper.The point-by-point response to the the reviewer’s comments is in the attachment.

Reviewer 3 Report
Review of manuscript entitled "Relationships between mineralized structures and induced polarization properties of seafloor polymetallic sulphide rocks: A numerical simulation study” by Wu et al., submitted to the journal MINERALS.
The above manuscript mainly discusses on the results of numerical simulation based investigation on the variability of induced polarization and electrical conductivity against variety of Polly-metallic sulphides having different mineral characters. Authors have attempted to project such imagining mechanism for various types of sulphide in host rocks including massive, disseminated and vein-type deposits and their characteristic polarization features as useful technique for commercial exploration of metallic sulphides from the seafloor. Thus the scientific concept presented in this article is quite interesting and also providing innovative information’s for future research.
However, the overall presentation of the manuscript still has several loopholes; which authors need to addressed/justify properly. Some of those are mentioned in the following section-
· At many parts of the text, the language used are not clear and in some cases, it is due to use of lengthy statements. To bring more clarity in the text, authors should avoid such long complex sentences.
· In the text, a couple of statements need to cite supportive references from available literatures. Particularly, the sentences in lines 45-47; 49-50; 137-138; 159-161; 297-299 etc.
· In methodology, a short description about the field based IP measurements in seafloor sulphide deposits also need to be incorporated (probably under the section 2.1). It would provide the reader a better understanding about the technical aspects of that method.
· Line 165: Its need to be mention about the method, that followed for estimation of actual background IP values due to rocks and minerals other than sulphides in such natural samples.
· In the Fig 3., the plot of numerical simulation-based data set of chargeability against volume of sulphide are compared with experimental data from Komori et al., 2017 and Chen et al., 2021. Those experimental data also showed considerable deviation from the simulated trend. What are the types of sulphides considered for those experimental results?
· In Tables 1,2, and 3, the simulation based results are presented for disseminated, vein type and massive sulphides. What are the sources and nature of those sulphides used for this study? Those details should be mentioned in short.
· In the caption of Table 1. remove the repeated statement.
· In Table 3, the lengths of longer axes are shorter than corresponding minor axes- This is bit confusing. What would be the aspect ratio for basalt? The units for those axis lengths are also missing in that table.
· The current density distributions for three different sulphide contents are presented in Figure 8. However, the actual value of different volume contents of disseminated, vein and massive sulphides (as represented by three columns’ in figs. 8a, 8b and 8c) are not mentioned in fig or its caption.
· Line 457: “……which is usually determined by the pore structure”. What is meant by “pore structure” in this sentence?
· Line 533: Instead of “…..sulphide when volume content is below 0.2.” use “……sulphide when volume content is below 20%.
· Line 533-534: “….total chargeability depends non-linearly on the volume content of vein and massive metal sulphide.” Author mentioned here about non-linear relations for vein and massive sulphides. But in Fig 3. the distributions of three points for vein sulphides (red stars) and massive sulphides (yellow circles) looks quite linear. Like disseminated sulphides, more points are necessary for vein and massive sulphide to establish such relation.
Author Response
Thank you for your suggestion and help. Those comments are all valuable and very helpful for revising and improving our paper, as well as the important guiding significance to our researches. According to your comments above, we tried our best to improve the manuscript and made some changes in the manuscript. The point-by-point response to the the reviewer’s comments is in the attachment.
